# Prospective of Agro-Waste Husks for Biogenic Synthesis of Polymeric-Based CeO_2_/NiO Nanocomposite Sensor for Determination of Mebeverine Hydrochloride

**DOI:** 10.3390/molecules28052095

**Published:** 2023-02-23

**Authors:** Gamal A. E. Mostafa, Maha F. El-Tohamy, Haitham Alrabiah

**Affiliations:** 1Department of Pharmaceutical Chemistry, College of Pharmacy, King Saud University, P.O. Box 2457, Riyadh 11451, Saudi Arabia; 2Department of Chemistry, College of Science, King Saud University, P.O. Box 22452, Riyadh 11495, Saudi Arabia

**Keywords:** mebeverine hydrochloride, metal oxides, nanomaterials, modified sensors, polymerized membranes, agro-husks

## Abstract

Background: The remarkable properties of nickel oxide (NiO) and cerium oxide (CeO_2_) nanostructures have attracted considerable interest in these nanocomposites as potential electroactive materials for sensor construction. Methods: The mebeverine hydrochloride (MBHCl) content of commercial formulations was determined in this study using a unique factionalized CeO_2_/NiO-nanocomposite-coated membrane sensor. Results: Mebeverine-phosphotungstate (MB-PT) was prepared by adding phosphotungstic acid to mebeverine hydrochloride and mixing with a polymeric matrix (polyvinyl chloride, PVC) and plasticizing agent *o*-nitrophenyl octyl ether. The new suggested sensor showed an excellent linear detection range of the selected analyte at 1.0 × 10^−8^–1.0 × 10^−2^ mol L^−1^ with regression equation E_mV_ = (−29.429 *±* 0.2) log [MB] + 347.86. However, the unfunctionalized sensor MB–PT displayed less linearity at 1.0 × 10^−5–^1.0 × 10^−2^ mol L^−1^ drug solution with regression equation E_mV_ = (−26.603 ± 0.5) log [MB] + 256.81. By considering a number of factors, the applicability and validity of the suggested potentiometric system were improved following the rules of analytical methodological requirements. Conclusion: The created potentiometric technique worked well for determining MB in bulk substance and in medical commercial samples.

## 1. Introduction 

Nanomaterials have recently been discovered as potential essential components in sensors, material development, medicinal applications, and cancer treatments targeting systems. In order to develop numerous sensitive sensors in our contemporary lives, scientists are working to broaden their field of study [1]. Numerous exciting characteristics of nanomaterials are being researched as potential answers to various modern problems. They offer a substantial new contribution to solving global and environmental challenges. The remarkable qualities of nanoparticles and nanomaterials make them ideal for creating sensing systems with improved performance [2]. They can be integrated into transducers by connecting to a suitably tailored substrate by changing their size and morphology, producing sensors with increased sensitivity and quicker response times [3]. Hybrid materials and nanocomposite structures comprised of metallic NPs, paired with certain conductive polymers and a modified electrode, have been created for electrochemical sensing due to their outstanding combination of biocompatibility, surface area, and conductivity [4].

Nickel oxide (NiO), is a significant transition metal oxide with a cubic lattice structure and has several applications [5], including electronic devices, thermal and magnetic materials, fuel cell catalysts, electrochromic films, and biomedicals [6,7,8]. Cerium oxide nanoparticles (CeO_2_NPs) have attracted a great deal of interest because of their outstanding catalytic activity. This has resulted from the rapid and efficient mutation of the oxidation state between Ce^4+^ and Ce^3+^, with potential applications in bioanalysis [9], biomedicine [10], drug delivery [11], and bio-scaffolding [12].

Agriculture has contributed significantly to the generation of waste and environmental contamination, just like other aspects of development activity [13]. Worldwide, the agricultural production system produces a lot of solid waste. Environmental degradation brought on by improper agro-waste management results in financial losses and issues with human health [14]. Therefore, it is vital to design and develop agro-waste management methods that are economical, practical, and socially acceptable. Literature surveys have described several approaches for the green synthesis of nanomaterials using agro-wastes and their application in various areas, including environmental applications [15], the elimination of antibiotics from aqueous solution [16], catalytic degradation [17], and biomedical applications, such as antimicrobial agents, drug targeting systems, and anticancer treatments [18]. The advantages of biogenic synthesis methods are that the costs of chemicals, reagents, and other additives are eliminated. Additionally, the environmental wastes are substantially reduced and can be used as reducing, capping, and stabilizing agents [19].

Mebeverine is a medication used to treat some of the signs and symptoms of irritable bowel syndrome (IBS) and associated diseases, particularly stomach cramps, recurrent diarrhea, and gas [20]. It does not interfere with the autonomic nervous system, thus, reducing spasms without impairing gastrointestinal motility. To prevent major health issues in the form of cardiovascular and neurological symptoms, the recommended mebeverine hydrochloride dosage is between 135 and 270 mg.

Different analytical techniques, such as chromatographic separation methods [21,22], spectroscopic [23,24] and electrochemical techniques [25,26,27,28,29], have been used to determine mebeverine hydrochloride. For a very long time, researchers from all over the world have preferred potentiometric sensors. Polymeric sensors can be used in many facets of daily life. They are employed in clinical diagnostics, the process of quality control, and environmental applications [30,31,32]. Their widespread use is due to the many benefits they provide, which include ease of use, short analysis times, affordable operating costs, low detection limits, and excellent selectivity [33].

The objective of this work was to create a modified metal oxide CeO_2_/NiO-nanocomposite-coated wire sensor with extremely high sensitivity and selectivity for MB detection in its commercial products. The sensitivity and selectivity of the potentiometric modified sensor were attempted to be improved using a novel approach that takes advantage of the special physical, chemical, optical, and electrical features of the chosen metal oxides. The inclusion of the proposed CeO_2_/NiO nanocomposite in the polymeric matrix was designed to influence the sensitivity and drug selectivity of the sensor. Method validation, performed in accordance with IUPAC recommendations [34], was undertaken as a means of determining whether the suggested method was analytically appropriate. Additionally, a comparison between the suggested CeO_2_/NiO-nanocomposite-coated membrane sensor and conventional membrane sensors was performed. 

## 2. Results and Discussion 

### 2.1. Characterization of the Synthesized Nanoparticles

Understanding the electrical nature and the optical band gap of the material may be assessed by a study of its UV-visible absorption spectrum properties. Electronic transitions inside the sample cause absorption in the near UV range. The optical studies of the pre-synthesized CeO_2_NPs, NiONPs and their nanocomposite were performed at ambient temperature in the wavelength range 200–500 nm (Figure 1). The UV-vis spectra were recorded after dispersion of the nanoparticles in deionized water and ultrasonicated for 10 min. A well-defined absorption peak can be seen at a wavelength of 310 nm for the CeO_2_NPs. The limited size distribution of ceria nanocrystals is well-matched with the acute and strong absorption spectra with a prominent excitonic feature. In contrast with the previously reported absorption peak of CeO_2_NPs at 321 nm [35], it was clear that the peak was shifted towards a lower wavelength, exhibiting blue shift (Figure 1a). This showed that the sizes and morphologies of CeO_2_ nanoparticles have an impact on the absorption positions. The UV-vis spectrum of NiONPs showed the presence of a distinct peak at 259 nm due to the electronic transition taking place between both valance and conductive bands (Figure 1c) [36]. However, the absorption spectrum of the CeO_2_/NiO nanocomposite showed an absorption band at 350 nm (Figure 1e). It is worth noting that, when CeO_2_ was loaded onto NiO, the absorption edges exhibited a red shift and an increase in absorption in both the UV and visible bands when compared to pure NiO and CeO_2_. In other words, the addition of CeO_2_ successfully extended the absorption to the visible range. Furthermore, the intensity of the absorption increased, indicating that the addition of CeO_2_ was favorable to improving the visible-light absorption capabilities of the CeO_2_/NiO heterojunctions [37]. The direct optical band gap of the synthesized CeO_2_NPs, NiONPs, and CeO_2_/NiO nanocomposite was estimated using the following equation: Energy Quantum Mechanics Eg = hC/λ(1)
where these values are expressed as Eg (band gap), h (bank constant, 6.626 × 10^−34^ J.s), C (velocity of light, 2.99 × 10^8^ m/s), and λ (absorption band value), respectively. The conversion factor 1 eV is equivalent to 1.6 × 10^−19^ J. The calculated band gaps were 3.26 eV, 4.78 eV, and 2.62 eV, for the above-described nanomaterials, respectively. The surface plasmon resonance (SPR), which also boosts the scattering probability and radiation penetration, gives the surface a reduction shape. The oxidation process is accelerated by these processes, which entail the surface-level formation of holes and the separation of electrons. Furthermore, it has been shown that changes to the dielectric matrix can influence the absorbance peak that can be seen on an SPR. The matrix’s effective dielectric behavior is known to be directly correlated with its refractive index.

The FT-IR analysis provides the possible functional groups that are present in the biomolecules and can be responsible for the reduction, capping and stabilization of the synthesized nanomaterials. The FT-IR spectrum of CeO_2_NPs (Figure 2a) showed a broad band around 3433 cm^−1^ reflecting the presence of the O-H stretching vibration of aromatic or aliphatic alcohols. The two observed bands at 2874 and 2355 cm^−1^ reflected the appearance of medium C-H stretching of alkane and strong O=C=O carbon dioxide stretching vibrations, respectively. The three bands observed at 1647, 1384, and 1103 cm^−1^ reflected the presence of strong H-O-H bending of the bonded water of cellulose crystals, asymmetric and symmetric deformation of C-H in alkane or phenolic alcohol, and the sugar units ester linkage, respectively. The absorption band at 1033 cm^−1^ was attributed to symmetric and asymmetric stretching of Si-O-Si and Si-O-C of silica present in the wheat husk extract. All the recorded peaks matched the previously reported phytochemical FT-IR spectrum of wheat husk extract [38]. The FT-IR band at 504 cm^−1^ was related to the formation of Ce-O [39]. 

Figure 2b displays the functional groups that formed during the synthesis of NiONPs. The starching bands at 3640 and 3431 cm^−1^ were attributed to sharp and broad O-H of Si-OH and the interlayer of water, respectively. The two bands appearing at 2918 and 2884 cm^−1^ showed the presence of weak intramolecular O-H attraction of alcohol and strong stretching of the C-H of alkane, respectively. The absorption bands at 2340, 1792, 1416, and 1076 cm^−1^ were due to the presence of O=C=O carbon dioxide, strong C=O acid anhydride, medium bending O-H of carboxylic acid, and the symmetric Si-O-Si group, respectively. The obtained bands were in agreement with the previously noted study of FT-IR of rice husk extract [40]. Furthermore, the observed peak at 436 cm^−1^ was attributed to Ni-O nanoparticles [41]. The FT-IR spectrum of CeO_2_/NiO nanocomposite showed the same absorption bands with a slight shift. The bands observed at 502 and 436 cm^−1^ showed the presence of Ce-O and Ni-O, respectively (Figure 2c).

Analytical methods such as XRD are used to identify and quantify various crystalline forms in samples. This investigation was conducted at (k = 1.5406 A) on a Cu-k XRD- diffractometer. The XRD diffraction study of CeO_2_NPs displayed different distinct diffraction peaks at 2θ values of 28.72° (1 1 1), 31.71° (2 0 0), 47.70° (2 2 0), 56.92° (3 1 1), 76.90° (331), and 79.2° (420), relating to the face-centered cubic CeO_2_ crystalline structure. These values were in agreement with those in the JCPDS card no.34-0394 for CeO_2_ (Figure 3a) [42]. The XRD spectrum of NiONPs (Figure 3b) showed sharp and significant peaks at 2θ = 37.26° (1 1 1), 43.50° (2 0 0), 63.08° (2 2 0), 75.39° (3 1 1), and 79.50° (2 2 2) crystalline planes, respectively. These diffraction results of NiONPs revealed the formation of a cubic crystalline phase of NiO and matched those reported in the JCPDS card for NiO 78-0643 [41]. 

The XRD pattern of CeO_2_/NiO nanocomposite showed three low intensity peaks, corresponding to 37.26° (1 1 1), 34.50° (2 0 0), and 63.08° (2 2 0) of NiO nanoparticles. However, various peaks were observed at 2θ values of 28.72° (1 1 1), 31.71° (2 0 0), 47.71° (2 2 0), 56.91° (3 1 1), and 76.90° (4 2 0), corresponding to CeO_2_NPs, revealing the successful formation of CeO_2_/NiO nanocomposite (Figure 3c). No other peaks appeared, indicating the high purity of the green-synthesized CeO_2_NPs and NiONPs, and their nanocomposite.

The Debye–Scherrer relation [43] was applied to estimate the average crystallite size in each green-synthesized nanomaterial.
Average crystallite size D = 0.94λ/βCosθ(2)
where D, λ, β, and θ are the average crystallite size, the absorption wavelength 1.54056 Å for Cu Kα radiation (a constant value), the width of the peak at half maximum intensity, and the peak position angle, respectively. The calculated values were found to be 25, 32, and 45 nm, for the CeO_2_NPs, NiONPs, and CeO_2_/NiO, respectively. 

To study the surface shape and size of the green-synthesized CeO_2_NPs, NiONPs, and the CeO_2_/NiO nanocomposite, SEM images were selected and are shown in Figure 4a–c. It can be seen that the synthesized CeO_2_NPs and NiONPs using wheat and rice husk extracts were morphologically sphere-shaped, with narrow particles indicating significant homogeneity of particle distributions and the formation of a vast mass of spherical agglomerates (Figure 4a,b). The SEM image of the nanocomposite showed condensed masses of spherical and agglomerated particles, confirming the formation of CeO_2_/NiO nanocomposite (Figure 4c). The particles recorded using SEM were found to be around 100 nm for the above three described nanomaterials.

An elemental analysis of the green-synthesized CeO_2_NPs, NiONPs, and CeO_2_/NiO nanocomposite was performed using SEM connected with an EDX spectrometer. With respect to the recorded results, it was found that, in the CeO_2_NPs sample, the weight% of Ce and O was 57.46% and 42.54%, while the atomic% was recorded to be 49.51% and 50.49%, respectively (Figure 5a). For the NiONPs sample, the calculated values were expressed as weight% 61.15% and 38.85% for Ni and O. Furthermore, the atomic% was observed to be 43.97% and 56.03% for Ni and O, respectively (Figure 5b). The EDX analysis of the CeO_2_/NiO nanocomposite exhibited weight % 36.38%, 31.02%, 32.6% for Ce, Ni, and O, respectively, and atomic % of 47.35%, 9.93%, and 42.72% for the three mentioned elements, respectively (Figure 5c).

EDX mapping of the pre-synthesized CeO_2_NPs, NiONPs, and CeO_2_/NiO nanocomposite are shown in Figure 6a–c. All the results obtained for SEM, EDX and elemental mapping confirmed the successful formation of a CeO_2_/NiO nanocomposite. 

### 2.2. The Response Features of the Designed Sensors

The potential response of the designed sensors was generated from the potential difference between the active sites (MB-PT or MB-PT-CeO_2_/NiO nanocomposite) and the MB ions in the analytical sample. The addition of *o*-NPOE as a plasticizer in the preparation of the coated membranes enhances the flexibility and durability of the membrane. Moreover, the presence of a plasticizer can act as an ionophore. This can be attributed to the donation sites and functional groups present in the plasticizer, which can affect the chelation of the primary analyte ion [44]. The characteristic behavior of the newly designed MB-PT and MB-PT-CeO_2_/NiO nanocomposite sensors was measured and is summarized in Table 1. With respect to the outcome data, the designed sensors showed Nernstian slopes of 26.603 ± 0.3 and 29.429 ± 0.2, covering 1.0 × 10^−5^ to 1.0 × 10^−2^ and 1.0 × 10^−8^ to 1.0 × 10^−2^ mol L^−1^ for the above-described sensors, respectively (Figure 7a,b).

To evaluate and determine the suitable independent pH range of the designed sensors, the effect of pH on the potential readings of MB-PT and MB-PT-CeO_2_/NiO sensors was tested, and the findings were plotted, as shown in Figure 8. It was observed that the independent pH range that can be used safely for the determination of MB was 4–8. At the acidic medium, the potential readings were gradually increased due to the competition between the MB analyte ions and the H^+^ ions. However, by increasing the alkalinity of the test solution the potential response of the sensors decreased gradually due to the lower solubility of the MB in the solution and the high concentration of OH^-^ in the test solution [45]. 

The newly designed MB-PT and MB-PT-CeO_2_/NiO nanocomposite sensors were evaluated using 50 mL of MB solution and foreign materials (1.0 × 10^−3^ mol L^−1^) separately, with respect to their selectivity towards the detection of MB^+^ ions. Various foreign materials, including cations (Ca^2+^, Mg^2+^, Cu^2+^, Zn^2+^, Na^+^, K^+^, Cr^3+^, and Al^3+^), amino acids (histidine and glycine), sugars (lactose and sucrose), and pharmaceutically formulated additives (talc and magnesium stearate) were tested. The obtained results exhibited excellent selectivity for functionalized MB-PT-CeO_2_/NiO nanocomposite. The addition of CeO_2_NPs and NiONPs enhanced the selectivity of the tested drug due to the smaller particle size and the distinct physical and chemical properties of the metal oxide nanoparticles. The free energy of MB^+^ transfer between the aqueous and active sites in the coated membrane is frequently used to characterize the selectivity of designed MB-PT and MB-PT-CeO_2_/NiO nanocomposite membrane sensors. No interference was recorded due to the difference in ionic size, mobility, and the permeability of interfering ions compared to MB^+^ (Table 2).

### 2.3. Quantification of Mebeverine Hydrochloride

The designed MB-PT and MB-PT-CeO_2_/NiO nanocomposite sensors were applied to quantify MB in its authentic form and the results were calculated as mean percentage recoveries. Table 3 presents the mean percentage recoveries as 98.88 ± 0.7 and 99.58 ± 0.4 for the above-described sensors, respectively. The use of metal oxide nanocomposite improved sensor stability in comparison to their bulk counterparts. Nanocomposites composed of CeO_2_NPs and NiONPs have exceptional physical and electrochemical performance. They are distinguished by their high surface-to-volume ratio and semiconducting properties. The presence of the metal oxide nanocomposite increases the electroactive surface area and enhances electron transport between the inner sensor and the ion-sensitive membrane.

### 2.4. Method Validation

The suitability and validity of the designed potentiometric systems were confirmed by following the previously recommended guidelines [34]. The designed sensors displayed linearity covering 1.0 × 10^−5^–1.0 × 10^−2^ and 1.0 × 10^−8^–1.0 × 10^−2^ mol L^−1^ for the MB-PT and MB-PT-CeO_2_/NiO nanocomposite sensors, respectively. The derived regression equations previously mentioned in Table 1 indicated the high sensitivity of the suggested sensor functionalized with metal oxide nanocomposite with a correlation coefficient r = 0.9999, compared to the MB-PT sensor of r = 0.9993. 

The low limit of detection (LOD) in the potentiometric assay could be estimated when the potential readings of the designed sensors dropped by 17.9 mV. The LOD was estimated to be 2.5 × 10^−6^ and 2.5 × 10^−9^ mol L^−1^ for the MB-PT and MB-PT-CeO_2_/NiO nanocomposite sensors, respectively. 

To ensure the accuracy of the developed technique, nine authentic MB sample solutions were analyzed using the proposed sensors. The accuracy was expressed as mean percentage recoveries (Mean ± SD). The findings were 99.65 ± 0.3% and 99.82 ± 0.2%, as shown in Table 4. 

Application of intermediate precision (intra-day and inter-day) assays to determine the degree of precision of the developed functionalized potentiometric system was performed. The estimated results were expressed as relative standard deviation percentage (RSD%). The findings were 0.4% and 0.2% for intra- and inter-day assays, respectively, revealing excellent precision (˂2%), as summarized in Table 5.

The robustness of the design system was determined by changing the pH to 8 ± 1 using phosphate; the findings were calculated to be 98.96 ± 0.8 and 99.87 ± 0.4% for the above two mentioned sensors, respectively. Furthermore, the reliability of the current technique was evaluated using another pH meter (Jenway-3510). The obtained data were 98.70 ± 0.6% and 99.74 ± 0.3% for the tested sensors. These results revealed no significant difference between the obtained results and those obtained from the analytical studies.

### 2.5. Quantification of the Drug in Tablets

Mebeverine hydrochloride was determined in its commercial product Duspatalin^®^200 mg/capsule using the MB-PT and MB-PT-CeO_2_/NiO nanocomposite sensors. The estimated results were found to be 98.72 ± 0.6 and 99.56 ± 0.4 for the two sensors, respectively (Table 6). The observed data indicated that the functionalized MB-PT-CeO_2_/NiO nanocomposite sensor exhibited more favorable sensitivity towards the detection of the MB analyte. The proper modification of the MB-PT-CeO_2_/NiO nanocomposite sensors showed effective detection of the target analyte due to the unique physical and chemical characteristics of the metal oxide nanocomposite, including the large surface area that enhances the interfacial contact between the active sites in the sensor membrane and the detected analyte ions, its high conductivity, and its dynamic stability. The obtained findings were statistically processed [46] and the estimated data compared with those obtained from a previously developed method based on the formation of a PVC sensor using silicotungstic acid [28]. The results were excellent and matched with no observed significant difference.

Further comparative evaluation was performed between the newly proposed sensors and three previous studies with respect to the type of reagent used, the linearity, and the lower limit of detection. 

Table 7 shows that the first study was based on the formation of a modified glassy carbon electrode using tricresylphosphate with a linearity of 3.0 × 10^−7^–1.0 × 10^−2^ and a detection limit of 3.0 × 10^−7^ mol L^−1^, whereas, the second report described the formation of polyvinyl chloride sensors using silicotungstic acid with a linearity range of 4.0 × 10^−6^–1.0 × 10^−2^ mol L^−1^. However, the third study used a liquid membrane electrode incorporating sodium tetraphenylborate. A lower detection range of 1.0 × 10^−5^–1.0 × 10^−1^ mol L^−1^ was reported using this method. The proposed method using functionalized MB-PT-CeO_2_/NiO nanocomposite sensors gave excellent sensitivity, selectivity, and fast dynamic response due to the previously mentioned extraordinary features of the nanomaterials. 

## 3. Experimental

### 3.1. Materials and Reagents

Two naturally prepared wheat (*Triticum aestivum)* and rice (*Oryza sativa*) husk extracts were used throughout this study to synthesize green metal oxide nanoparticles (cerium oxide and nickel oxide nanoparticles, CeO_2_NPs and NiONPs, respectively). All pure grade materials and reagents were purchased from Sigma-Aldrich (Hamburg, Germany), including cerium nitrate hexahydrate (99.9%), nickel nitrate hexahydrate (99.9%), phosphotungstic acid hydrate (99.9%), polymeric material (polyvinyl chloride, PVC), fluidizing materials (*o*-nitrophenyl octyl ether, *o*-NPOE), as well as other solvents, such as tetrahydrofuran (THF, 97.0%), methanol (99.9%), ethanol (99.9%), and acetone (99.9%). Pure grade mebeverine hydrochloride was supplied from Sigma-Aldrich (Chemi GmbH, Germany). The pharmaceutical product Duspatalin^®^ capsules (200 mg/capsule) was purchased from a local pharmacy (Al-nahdi pharmacy, Riyadh, Saudi Arabia). 

### 3.2. Instrumentation 

The experimental measurements were carried out using HANNA-211 (Smithfield, RI, USA). The pH optimization was achieved by another pH meter (Metrohm-744, (Herisau, Switzerland). The characterization of the synthesized nanomaterials was performed using a Perkin-Elmer BX spectrometer (Waltham, MA, USA), a Shimadzu X-ray-6000 diffractometer (Kyoto, Japan), JSM-7610, and a scanning electron microscope (SEM, Tokyo, Japan), connected with energy-dispersive X-ray spectroscopy (EDX, Tokyo, Japan).

### 3.3. Eco-Friendly Synthesis of Nanomaterials 

#### 3.3.1. Preparation of Wheat (*Triticum aestivum*) and Rice (*Oryza sativa*) Husk Extracts

The husks of wheat and rice were collected separately and sieved to remove any tainted materials. Then, six times distilled water was used to wash out any remaining contaminants from the collected husks. The cleaned husks were dried for 24 h at 70 °C in a hot air oven. About 30 g of the powdered dried husks (50–200 µm) was heated in 300 mL of distilled water at 60 °C for two hours under continuous magnetic stirring. Then a Whatman filter paper No. 40 was used for the filtration process. The extracted material was kept in an Erlenmeyer flask at 4 °C to be used later in the production of the metal oxide nanomaterials. Then the material was filtered using a filter (Whatman paper No. 40). The obtained extract of each husk was stored in an Erlenmeyer flask at 4 °C for further use in the synthesis of metal oxide nanomaterials. 

#### 3.3.2. Synthesis of CeO_2_NPs and NiO Nanoparticles Using Husk Extracts

The phytocomponents of wheat and rice husk extracts, such as cellulose, hemicellulose, lignin, starch, proteins and fatty acids, act as reducing as well as stabilizing agents [47]. To synthesize CeO_2_NPs and NiONPs, approximately 50 mL of wheat *(Triticum aestivum)* and rice *(Oryza sativa)* husk extracts was heated separately to 60 °C for 30 min; then 50 mL of 1.0 mol L^−1^ of cerium nitrate hexahydrate and nickel nitrate hexahydrate was added to the heated extracts, respectively. A few drops of 2.0 mol L^−1^ sodium hydroxide were added dropwise to the above mixtures and continuously heated at 80 °C for 2 h under constant stirring until the formation of CeO_2_NPs and NiONPs. The formed nanoparticles were washed three times using deionized water to remove excess sodium hydroxide. The obtained nanoparticles were then filtered using Whatman filter paper No.1 and oven dried at 100 °C for 4 h. The synthesis of nanomaterials is illustrated in Figure 1.

### 3.4. Preparation of Analytical Solutions 

Serial dilution of MB drugs in the concentration range of 1.0 × 10^−8^–1.0 × 10^−2^ mol L^−1^ using distilled water was performed from aqueous 1.0 × 10^−2^ mol L^−1^ of MB drug solution (0.466 g/100 mL *w*/*v*).

### 3.5. Preparation of Electroactive Material

The electroactive material MB-PT was obtained by mixing equal volumes and concentrations of each MB drug solution and the precipitating agent PTA (50 mL, 1.0 × 10^−2^ mol L^−1^). The resulting electroactive material MB-PT was washed 3 times with distilled water, collected by filtration using Whatman filter paper No.1 and kept aside overnight at ambient temperature for drying.

### 3.6. Sensor Preparation 

The membrane cocktail of MB-PT and MB-PT-CeO_2_/NiO nanocomposite was formed by mixing PVC (190 mg), MB-PT (10 mg), and *o*-NPOE (0.4 mL) with the organic solvent THF (5 mL). The blended mixture was poured into a rounded 3 cm diameter Petri dish and allowed to dry at ambient temperature until the formation of an oily membrane cocktail. A pure Al wire was polished and washed with acetone 3 times and then dipped in the previously prepared membrane cocktail to fabricate the coated sensor MB-PT. The functionalized mixture was prepared by the same above-mentioned procedure but approximately 10 mg of CeO_2_/NiO nanocomposite was added to the membrane cocktail under magnetic stirring for 10–15 min to obtain a homogeneous dispersed membrane mixture. After coating the Al wire using the functionalized mixture, the sensors were hung to dry overnight at room MB-PT-CeO_2_/NiO nanocomposite sensor temperature, then preconditioned by soaking the fabricated sensors in 1.0 × 10^−10^ mol L^−1^ MB solution for 12 h. Figure 2 describes the potentiometric system using a modified MB-PT-CeO_2_/NiO nanocomposite sensor.

### 3.7. Calibration Plots 

The linear concentration range of the suggested new sensors can be graphed by plotting the potential readings (mV) of the sensors vs. -logarithm of the MB concentrations using the designed indicator sensor and the reference (Ag/AgCl) electrode. The tested MB solutions were 50 mL each of the concentration range 1.0 × 10^−8^ to 1.0 × 10^−2^ mol L^−1^.

### 3.8. Optimization of Analytical Conditions

Various factors that affect the potential readings of sensors should be investigated and optimized, including the pH, selectivity, and response time. The pH of 1.0 × 10^−4^ mol L^−1^ of MB solution (50 mL) was tested using the designed sensors by changing the pH from acidic to alkaline values. The pH of the test solution was acidified using a few drops of 0.1 mol L^−1^ of hydrochloric acid and then elevated using the same concentration of sodium hydroxide. A combined glass electrode was conjugated with a potentiometric system and the pH values vs. the potential readings (mV) were plotted to estimate the independent pH values for the designed sensor [48]. 

The selectivity of the applied sensor in analytical detection is the most important factor which may affect the potential response of the sensor. The selectivity coefficient was determined to see how much a foreign substance would influence the sensor response to its main ion. The selectivity coefficients were then determined using the separate solutions method (SSM) in accordance with IUPAC rules [34].

Briefly, the selectivity coefficient $\;K_{MB}^{pot}$ was estimated by measuring the potential readings of the sensor in 1.0 × 10^−3^ mol L^−1^ of each MB solution and possible foreign substances separately, such as particular cations, sugars, and polysaccharides, amino acids, and coformulation additives. The tolerable values were calculated using the previously reported equation [49].
Log Kpot = (E2 − E1)/S + Log [Drug] − Log [B^Z+^]1/z(3)
where Kpot represents the (selectivity coefficient), E1 is the electrode potential of 1.0 × 10^−3^ mol L^−1^ MB, E2 is the electrode potential of interfering species, B^z+^ is the interfering species, and S is the slope of the calibration graph.

The response time was estimated by recording the potential response of 1.0 × 10^−8^ to 1.0 × 10^−2^ mol L^−1^ of the MB solutions.

### 3.9. Analytical Determination of MB in Dosage Form 

The content of ten Duspatalin^®^ capsules (200 mg MB/capsule) was collected and weighed. A standard solution of 1.0 × 10^−2^ mol L^−1^ MB (0.466 g/100 mL *w*/*v*) was prepared in distilled water. The desired analytical samples were prepared by diluting the previously prepared standard to obtain 1.0 × 10^−9^ to 1.0 × 10^−2^ mol L^−1^ using distilled water. Each MB concentration was measured separately using the designed potentiometric systems MB-PT and MB-PT-CeO_2_/NiO nanocomposite; the regression equation was used to estimate the nominal content of MB.

## 4. Conclusions

The current study proposed MB-PT and MB-PT-CeO_2_/NiO nanocomposite sensors for the determination of mebeverine hydrochloride in authentic and commercial products. The advanced and unique characteristics of the functionalized metal oxide CeO_2_/NiO-nanocomposite-coated membrane sensors displayed excellent detection of MB with respect to the percentage recoveries, linear concentration range, precision, and the limit of detection. When comparing the potential readings of modified sensors to those of regular sensors in terms of sensitivity and selectivity, the created sensors performed better than existing traditional sensors. Additionally, the measurement of the chosen medication was carried out with good selectivity and over a broad linear concentration range with a low limit of detection due to the use of coated-membrane modifiers with metal oxide nanoparticles. The metal-oxide-enhanced membrane sensors can, therefore, be used by pharmaceutical companies, hospitals, and research laboratories to monitor pantoprazole sodium on a regular basis.

## Data Availability

All data involved in this study are included within the text.

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
