# Peer review of "Prospective of Agro-Waste Husks for Biogenic Synthesis of Polymeric-Based CeO2/NiO Nanocomposite Sensor for Determination of Mebeverine Hydrochloride"

_molecules, 2023, doi:10.3390/molecules28052095_

Round 1

Reviewer 1 Report

This submitted manuscript described a method using two naturally prepared wheat (Triticum aestivum) and rice (Oryza sativa) husk extracts to synthesize green metal oxide nanoparticles (Cerium oxide and nickel oxide nanoparticles (CeO2NPs and NiONPs). They were later used to form a unique factionalized CeO2/NiO nanocomposite-coated membrane sensor for Mebeverine Hydrochloride detection. Adequate methods such as FT-IR, SEM, XRD, EDS, etc., were used to demonstrate the successful synthesis of the material. The unfunctional sensor MB-PT displayed less linearity than the new suggested MB-PT-CeO2/NiO nanocomposite sensor, and both of them were applied to determine the Mebeverine Hydrochloride in tablets. The works in this manuscript was sufficient, and some suggestions are as follows:

1. What is the role of the two naturally prepared wheat and rice husk for CeO2/NiO nanocomposite, reducing, capping, or stabilizing agents?

2. Page 5 line 204, it is recommended to list the detailed equation.

3. The horizontal axis of the graph is incorrectly represented and should be logarithmic in Fig.7.

4. Table 1, please specify the meaning of intercept.

5. Scheme 2, the pH should be replaced with an instrument that measures potential.

6. Line 236 of page 6. The calculated band gaps were exhibited as 3.26 eV, 3.87 eV, and 2.62 eV... The 3.87 eV is inconsistent with the Eg value 4.78 eV of Figure 1d, please confirm.

7. Line 318 of page 10. while the atomic% was recorded to be 49.51 (Figure 5a). Please correct the data.

8. Line 321 of page 11. The EDX analysis of the CeO2/NiO nanocomposite exhibited weight% 36.38%, 31.02%, 32.6% for Ce, Ni, and O, respectively and atomic% of 47.35%, 9.93%, and 47.35% for the three mentioned element, respectively (Figure 5c). These data are inconsistent with Figure 5c, please correct.

9. Figure 3 is not labeled with the number a,b,c of the image.

10. Line 377, BM should be changed to MB.

11. What is the calculation of the selectivity factor ?  What are the units of the values in Table 2?

12. In Figure 1e, why does the UV-vis absorption spectrum of CeO2/NiO nanocomposite have only one peak instead of two, and what kind of combination or chemical reaction occurs between CeO2 and NiO?

13. Please check that the formatting of the references.

Author Response

Manuscript ID: molecules-2213780

Thank you very much for your valuable comments. The manuscript has been carefully revised

and all points have been corrected. The corrected points have been highlighted using red color.

  1. What is the role of the two naturally prepared wheat and rice husk for CeO2/NiO nanocomposite, reducing, capping, or stabilizing agents?

Answer: The role of wheat and rice husk was added in the text lines143-145.

  1. Page 5 line 204, it is recommended to list the detailed equation.

Answer: Page 5 line 204 the detailed equation has been added in the text.

  1. The horizontal axis of the graph is incorrectly represented and should be logarithmic in Fig.7.

Answer: Figure 7 has been corrected, (-log [MB], mol/L)

  1. Table 1, please specify the meaning of intercept.

Answer: The intercept in table 1 has been specified

  1. Scheme 2, the pH should be replaced with an instrument that measures potential.

Answer: Scheme 2, pH meter has been replaced by the instrument that measure potential.

  1. Line 236 of page 6. “The calculated band gaps were exhibited as 3.26 eV, 3.87 eV, and 2.62 eV...” The 3.87 eV is inconsistent with the Eg value 4.78 eV of Figure 1d, please confirm.

Answer: The band gap 3.87 has been corrected in the text to be 4.78 (line 244)

  1. Line 318 of page 10. “while the atomic% was recorded to be 49.51 (Figure 5a).” Please correct the data.

Answer: Figure 5a, the data has been corrected in the text (Line 326).

  1. Line 321 of page 11. “The EDX analysis of the CeO2/NiO nanocomposite exhibited weight% 36.38%, 31.02%, 32.6% for Ce, Ni, and O, respectively and atomic% of 47.35%, 9.93%, and 47.35% for the three mentioned element, respectively (Figure 5c).” These data are inconsistent with Figure 5c, please correct.

Answer: Figure 5 has been revised and the data of Figure 5c has been corrected in the text. 

  1. Figure 3 is not labelled with the number a,b,c of the image.

Answer: Figure 3 has been revised and labeled using a,b,c

  1. Line 377, BM should be changed to MB.

Answer: In line 377 BM has been corrected to be MB

  1. What is the calculation of the selectivity factor?  What are the units of the values in Table 2?

Answer: The selectivity or the interference of the applied sensor in analytical detection is the most important factor which  show the effect of other species on the potential response of the target analyte. The selectivity coefficient was determined to see how much a foreign substance would influence the sensor response to their main ion. The selectivity coefficients were then determined using the separate solutions method (SSM) in accordance with IUPAC rules [43].  (Lines 200-203)

  1. In Figure 1e, why does the UV-vis absorption spectrum of CeO2/NiO nanocomposite have only one peak instead of two, and what kind of combination or chemical reaction occurs between CeO2and NiO?

Answer:

It was worth noting that when CeO2 was loaded onto NiO, the absorption edges exhibited a red shift and an increase in absorption in both the UV and visible bands when compared to pure NiO and CeO2. In other words, the addition of CeO2 successfully extended the absorption to the visible range. Furthermore, the intensity of the absorption increased, indicating that the addition of CeO2 was favorable to improving the visible-light absorption capabilities of the CeO2/NiO heterojunctions. (This explanation has been added in the text with ref.40)

  1. Please check that the formatting of the references.

Answer: The references have been revised according to the Journal style. 

On behalf of my co-authors, I trust that our responses have addressed all of the concerns expressed and look forward to the acceptance of our revised manuscript. Prof. Dr. Gamal Mostafa

Prof. Dr. Gamal Mostafa

Reviewer 2 Report

The manuscript describes a development of a sensor based on nanocomposite produced by agro-waste husks (rice and wheat). The authors present comparative data with a sensor without nanocomposite. Below are some contributions to improve the manuscript:

1. Some acronyms are not defined in the text, such as PNZ.

2. Table 4 and 5 shown the accuracy results. Although the results are interesting, it is not possible to understand the concentration unit of the standards used. The same to Table 6.

3. I suggest that more information about selectivity studies be added in the text. It is not clear what the value obtained means in terms of selectivity.

Author Response

Manuscript ID: molecules-2213780

Thank you very much for your valuable comments. The manuscript has been carefully revised

and all points have been corrected. The corrected points have been highlighted using red color.

  1. Some acronyms are not defined in the text, such as PNZ

Answer: The title of Figure 7 has been corrected.

  1. Table 4 and 5 shown the accuracy results. Although the results are interesting, it is not possible to understand the concentration unit of the standards used. The same to Table 6.

Answer: The unit has been corrected and added in all tables.

  1. I suggest that more information about selectivity studies be added in the text. It is not clear what the value obtained means in terms of selectivity.

Answer: The selectivity of the applied sensor in analytical detection is the most important factor which may affect the potential response of the sensor. The selectivity coefficient was determined to see how much a foreign substance would influence the sensor response to their main ion. The selectivity coefficients were then determined in using the separate solutions method (SSM) in accordance with IUPAC rules [43].  (Lines 200-203)

On behalf of my co-authors, I trust that our responses have addressed all of the concerns expressed and look forward to the acceptance of our revised manuscript. Prof. Dr. Gamal Mostafa

Prof. Dr. Gamal Mostafa

Reviewer 3 Report

Ref: molecules-2213780

Title of the manuscript:Prospective of agro-waste husks for biogenic synthesis of polymeric-based CeO2/NiO Nanocomposite Sensor for Determination of Mebeverine Hydrochloride.”

 In this paper, authors designed an agro-waste based nanocomposite sensor for Mebeverine Hydrochloride. The novelty of the paper is missing. The paper is quite long and there is no interest to the readers. The manuscript should be rewritten in form of continuity. The crux of the research should be more highlighted and crisper than elongated details about potentiometric sensors and nanoparticles. Overall, I find this paper's results to be fine but the manuscript is suggested to be rewritten with improved English. 

Author Response

Manuscript ID: molecules-2213780

Thank you very much for your valuable comments. The manuscript has been carefully revised

and all points have been corrected. The corrected points have been highlighted using red color.

Title of the manuscript: “Prospective of agro-waste husks for biogenic synthesis of polymeric-based CeO2/NiO Nanocomposite Sensor for Determination of Mebeverine Hydrochloride.”

 In this paper, authors designed an agro-waste based nanocomposite sensor for Mebeverine Hydrochloride. The novelty of the paper is missing. The paper is quite long and there is no interest to the readers. The manuscript should be rewritten in form of continuity. The crux of the research should be more highlighted and crisper than elongated details about potentiometric sensors and nanoparticles. Overall, I find this paper's results to be fine but the manuscript is suggested to be rewritten with improved English. 

Answer: The manuscript has been revised carefully and all points suggested by reviewer have been corrected point by point and highlighted in the text using red colour. The introduction section rewritten again to be more interest to the reader. The novelty of this study has been added in the end of the introduction section. The manuscript has been revised with respect The English and it has been improved.

On behalf of my co-authors, I trust that our responses have addressed all of the concerns expressed and look forward to the acceptance of our revised manuscript. Prof. Dr. Gamal Mostafa

Prof. Dr. Gamal Mostafa

Round 2

Reviewer 1 Report

No more comments.

Reviewer 3 Report

Ref: molecules-2213780

Title of the manuscript:Prospective of agro-waste husks for biogenic synthesis of polymeric-based CeO2/NiO Nanocomposite Sensor for Determination of Mebeverine Hydrochloride.

In this paper, Mostafa et. al designed an agro-waste-based nanocomposite sensor for Mebeverine Hydrochloride. The authors have revised the manuscript quite well. The manuscript can be considered for publication.

1.      Manuscript should be checked thoroughly for English Grammatical errors like Line 45, “include” should be written as “including”; ‘fuel cells” should be written as “fuel cell”; etc.